# Quasi Non-Destructive Quality Assessment of Thermally Sprayed AISI 316L Coatings Using Polarization Measurements in 3.5% NaCl Gel Electrolyte

Maximilian Grimm [1,*], Pia Kutschmann [1], Christian Pluta [2], Olga Schwabe [3], Thomas Lindner [1] and Thomas Lampke [1]

1   Materials and Surface Engineering, Institute of Materials Science and Engineering, Chemnitz University of Technology, D-09107 Chemnitz, Germany
2   JELN Imprägnierung GmbH, D-41366 Schwalmtal, Germany
3   Putzier Oberflächentechnik GmbH, D-42799 Leichlingen, Germany
*   Correspondence: maximilian.grimm@mb.tu-chemnitz.de

**Abstract:** There is currently a lack of suitable methods of non-destructive quality assessment of thermally sprayed coatings. Therefore, this study investigates the suitability of polarization measurements that are adapted to the special needs of thermally sprayed coatings for non-destructive quality testing. For this purpose, a gel electrolyte containing 3.5% NaCl and a measuring cell based on the three-electrode arrangement were developed to prevent the corrosion medium from infiltrating the typical microstructure of thermally sprayed coatings (pores and microcracks). The newly developed method was evaluated on AISI 316L coatings deposited by high velocity air fuel (HVAF) and atmospheric plasma spraying (APS). The polarization curves showed significant differences as a result of spraying process-related changes in the coating microstructure. Even slight differences in oxide content within the AISI 316L coating produced by APS can be detected by the new method. In order to verify the new findings, the coatings were analyzed regarding their microstructure by optical microscope, scanning electron microscope and energy dispersive X-ray spectroscopy. The measuring cell and gel electrolyte developed offer a promising opportunity to evaluate the quality of thermally sprayed coatings in a largely non-destructive manner using polarization curves.

**Keywords:** quality control; thermally sprayed coatings; AISI 316L; polarization measurement; corrosion

## 1. Introduction

The property profile of thermally sprayed coatings depends to a large extent on their microstructure. Both a variety of adjustable parameters in the thermal spraying process used as well as some uncontrollable factors during the deposition process strongly influence the resulting coating microstructure and thus also the properties in service [1–3]. For the optimization of thermal spray processes as well as for the quality control of coatings, detailed microstructure analyses are of great importance. For this purpose, destructive examination methods, such as the metallographic preparation of transverse sections in order to evaluate the microstructure in the optical or scanning electron microscope, are often used. This type of examination provides substantial results, but is often costly and time-consuming, especially when dealing with single pieces or small series.

Electrochemical-instrumented tests (e.g., polarization curves or electrochemical impedance spectroscopy (EIS)) are used to characterize the corrosion behavior under specific conditions and have a number of advantages over other corrosion test methods that make electrochemical-instrumented tests interesting for use in quality assurance. Compared to other corrosion tests such as Corrodkote and neutral salt spray tests, electrochemical-instrumented tests are less time-consuming and show a high sensitivity to microstructural

changes as well as a high reproducibility [4–7]. The papers cited in the following are representative of how even minor microstructural changes affect the corrosion behavior and can be detected using polarization curves. Bastos et al. [5] shows that precipitation and phase transformation in the superduplex stainless steel UNS S32750 lead to a reduction in pitting potential and an increase in corrosion current density. Calliari et al. [6] stated that the formation of carbide precipitates in heat-treated Ni-Mo martensitic stainless steel causes an increase in the passivation current density in the polarization curves. Pacquentin et al. [7] studied the influence of changes in the microstructure of AISI 304L stainless steel induced by surface melting using a nano-pulsed laser on polarization curves. They revealed that the disappearance of the cold-rolled surface and the transformation of the martensitic phase into the high-temperature δ-ferritic phase leads to a higher breakthrough potential.

Although polarization measurements are also frequently conducted on thermally sprayed coatings, due to the use of aqueous electrolytes in combination with the typical structure of these coatings, which is often characterized by pores and microcracks, such measurements must be considered in a critical context. Depending on the coating microstructure, the aqueous electrolyte can infiltrate the coating. This leads to an out-of-control increase of the real measurement area, which results in a systematic error when determining important characteristic values such as the corrosion current density. Therefore, the measurements are often influenced by the substrate material used to such an extent that the corrosion behavior of the coating can no longer be accurately assessed [8–10]. However, even if the infiltration of the coating does not extend to the coating–substrate interface, slight effects such as those caused by minor microstructural changes can hardly be detected because of the undefined measurement area.

In the last decade, corrosion measurement methods using gel electrolytes have emerged in corrosion science, focusing on visualization of corrosion processes and on-site corrosion testing [11–19]. The results of the studies show that gel electrolytes provide corrosion characteristics similar to aqueous electrolytes and are therefore suitable for replacement. Up to now, gel electrolytes have been used with the aim of performing mobile measurements that are independent of laboratory scale, as required, for example, for characterizing the surfaces of cultural heritage [20,21]. The very high viscosity of the gel electrolytes enables easy handling and the tightness of the measuring system. Another reason for using gel electrolytes is to visualize corrosion processes in order to identify the location of the corrosion attack. For this purpose, various indicators are added to the gel electrolyte which mark the occurrence of corrosion products by a color change. Due to the limited mobility of the corrosion products in the gel electrolyte, the discoloration occurs at the site of the corrosion attack. The suitability of electrochemical measurements with gel electrolytes for the determination of microstructural differences in thermally sprayed coatings has scarcely been investigated yet.

Therefore, this study investigated to what extent the corrosion current density determined from polarization curves can be used to detect microstructural changes in AISI 316L coatings deposited using a high velocity air fuel (HVAF) and atmospheric plasma spraying (APS) process. For the measurements, both a 3.5% NaCl gel electrolyte and an adapted measuring cell based on the three-electrode arrangement were developed. An earlier study [8] already proves that the measurement setup is suitable to avoid an influence of the substrate on the polarization curve in contrast to the use of aqueous electrolytes and provides a comparison of the test method described here and the conventional one (aqueous electrolyte, Ag/AgCl sat. KCl as reference electrode). In later stages of development, the measuring cell should enable short-time corrosion measurements directly on the component surface in order to use electrochemical characteristics for non-destructive quality assessment of thermal spray coatings.

## 2. Materials and Methods

To investigate the effect of some coating microstructure characteristics on the polarization curve, low-carbon steel (EN 1.0117) plates ($100 \times 100 \times 4$ mm$^3$) were coated using

a commercially available gas-atomized AISI 316L powder in HVAF and APS processes. Before deposition, the substrates were grit-blasted and cleaned with a strong compressed air flow to ensure good bonding. The coatings were applied with a HVAF M3TM equipment (Uniquecoat Technologies, LLC, Oilville, VA, USA) using a propane-air mixture for combustion and nitrogen as powder carrier gas and a 309 MB plasma torch (GTV GmbH, Luckenbach, Germany) using an argon–hydrogen mixture for plasma generation and nitrogen as powder carrier gas. Each coated plate was cut into 25 samples ($20 \times 20 \times 4$ mm$^3$) on which the corrosion tests were performed. All samples were ground prior testing to provide a comparable surface condition.

For the measurement of the open circuit potential (OCP) and the polarization curve, a measuring cell and gel electrolyte was developed. The cell illustrated in Figure 1 is based on the conventional three-electrode arrangement, where the sample acts as the working electrode (WE) and a platinum sheet ($20 \times 20$ mm$^2$) as the counter electrode (CE). A thin platinum wire (Ø 0.5 mm) is used as pseudo-reference electrode (G-RE) and is positioned a few 100 µm in front of the sample surface. The pseudo reference is insulated by a sheath so that only the small cross section opposite the working electrode is in contact with the gel electrolyte. The measuring area has a diameter of 10 mm. The modified setup of the measuring cell shown in Figure 1b represents a significant simplification to the conventional three-electrode setup with Haber–Luggin capillary and reference electrode and is intended to facilitate easy handling for mobile measurements directly on the component surface. In addition to water, the gel electrolyte contains a 30% proportion of technical gelatin and some minor additives to prevent bubble formation and increase storage life. The NaCl concentration of the electrolyte is 3.5 wt %. To produce the gel, the technical gelatin and the additives are added to the heated, aqueous salt solution (approx. 50 °C) with constant stirring. After complete dissolution of all components, solidification to the gel takes place during cooling.

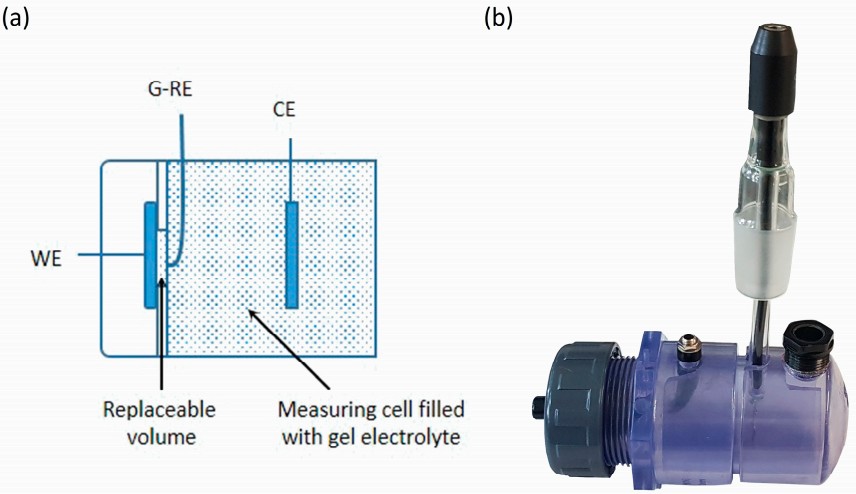

**Figure 1.** Schematic setup (**a**) and image (**b**) of the three-electrode arrangement for polarization measurements in gel electrolyte with WE = sample, G-RE = platinum wire, and CE = platinum sheet.

The tests were performed at room temperature using a potentiometer Zennium (Zahner-Elektrik GmbH & Co. KG, Kornach, Germany) and the ThalesZ software package (version 5.6.0 USB) (Zahner Elektrik GmbH & Co. KG, Kronach, Germany). The polarization curves were evaluated using a self-provided MatLab script determining $i_{corr}$ and $E_{corr}$ based on the Butler–Volmer equation. Before starting the potentiodynamic polarization scan, the OCP was measured for 15 min. Immediately afterwards, the samples were polarized with a scan rate of 1 mV/s in a range from –100 mV to +300 mV relative to the OCP. It should be noted that a lower scan rate is advisable for accurate assessment of the reaction kinematics. Due to the aim of the study using the measurement as a fast evaluation method

of the coating quality, trade-offs were made in favor of the measurement duration. With only a low polarization range ($-100$ mV to $+300$ mV relative to the OCP), the corrosion attack is hardly detectable visually and is limited to the immediate surface area (<10 μm). In order to determine the location and type of corrosion attack, the polarization range was increased for some samples ($-100$ mV to 700 mV relative to the OCP). As a result of the corrosion attack of the coating during the measurement, a small gel volume in front of the working electrode is contaminated as shown in Figure 2. Therefore, before each measurement, this gel volume is replaced by filling in new gel in a warm (about 40 °C), highly viscous state (approx. 2000 mPa·s). The small volume and only slightly increased temperature ensure very rapid cooling and solidification of the gel electrolyte to avoid infiltration of the coating. All samples were measured in a random order, to exclude all possible influences (e.g., aging effects of the gel electrolyte remaining in the measuring cell) of the corrosion measurement method.

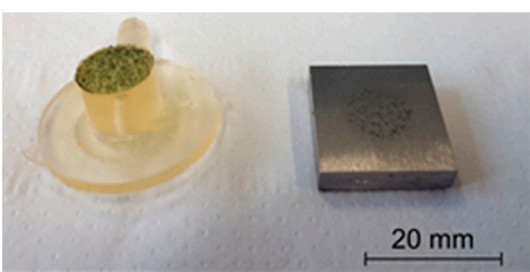

**Figure 2.** Intensified corrosion attack on 316L coated sample and resulting contamination of the gel electrolyte after polarization measurement with high polarization range (up to +700 mV relative to OCP). Replacement of a small volume of gel electrolyte is sufficient due to the localized contamination by corrosion products.

Cross sections were prepared by standard metallographic procedures to characterize the coating microstructure and corrosion phenomena. The porosity of the coatings was evaluated from five images taken with a GX51 optical microscope (Olympus, Shinjuku, Japan) equipped with a SC50 camera (Olympus, Shinjuku, Japan), using an image analysis method provided by the camera software. A SEM (LEO 1455VP, Zeiss, Oberkochen, Germany) operating with an accelerating voltage of 25 kV was used for more detailed investigations. Therefore, the secondary electrons (SE) were used to visualize the topographical differences resulting from the corrosion attack. Using the backscattered electron detector (BSD), differences in chemical composition (e.g., oxidized areas) become obvious. In addition, the chemical composition of the coatings, both average and locally, was determined by EDS analysis (GENESIS, EDAX, Mahwah, NJ, USA). To determine the average coating composition, large areas of the coating ($300 \times 100$ μm$^2$) were measured at three randomly selected locations. The local chemical composition of representative splats was studied at five measuring points. The phase composition was studied by X-ray diffraction (XRD) (D8 Discover diffractometer, Bruker AXS, Billerica, MA, USA) using Co Kα radiation with a tube voltage of 40 kV and a tube current of 40 mA. The diffraction patterns were measured for a 2θ range from 20° to 130°, with a step size of 0.01° and a dwell time of 1.5 s/step.

## 3. Results and Discussion

Figure 3 shows the AISI 316L coated plates prepared by HVAF and APS. While the coating sprayed with HVAF exhibits a uniform coloration, the APS coating presents significant differences in coloration. Only in the upper row, the coloration of the APS sample is close to that of the HVAF-coated plate and turns darker towards the lower row. A dark discoloration in thermally sprayed coatings is generally associated with increased oxidation of the particles or splats. It can be assumed that the higher thermal energy occurring in the APS process leads to a considerable temperature rise in the substrate and,

thus, also in the coating, with increasing deposition time promoting post-impact oxidation of the splats.

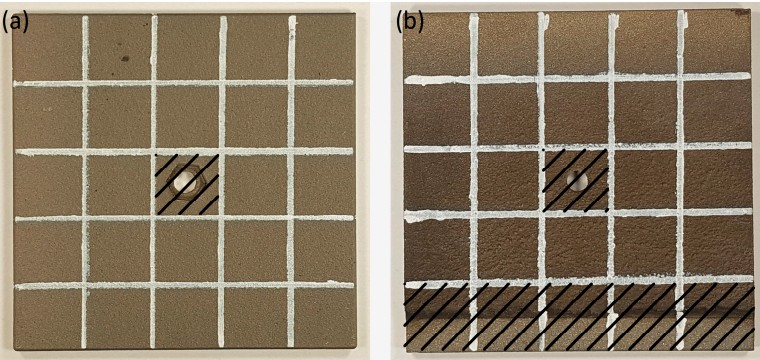

**Figure 3.** Images of the coated plates: (**a**) HVAF—uniform colouration, (**b**) APS—dark discolouration indicates more oxidation. Striped areas are not considered for measurement due to incomplete deposition caused by the mounting of the plate.

The SEM images in Figure 4 present the microstructures of the coatings investigated. Due to the characteristics of the spraying processes used (HVAF, APS), the coating microstructures differ significantly. As a result of the lower particle temperatures and higher velocities in the HVAF spray process, the shape of the powder particles is still visible in this type of coating. In addition to a few pores, dark oxide-rich areas mainly occur at the particle boundaries. The high particle temperatures occurring in the APS process result in a typical lamellar splat structure. In addition to pores, these coatings contain larger amounts of oxides. They are present as an accumulation in small areas or as completely oxidized splats. The average porosity of the coatings is similar to each other (HVAF: $(1.3 \pm 0.2)\%$, APS: $(1.8 \pm 0.5)\%$).

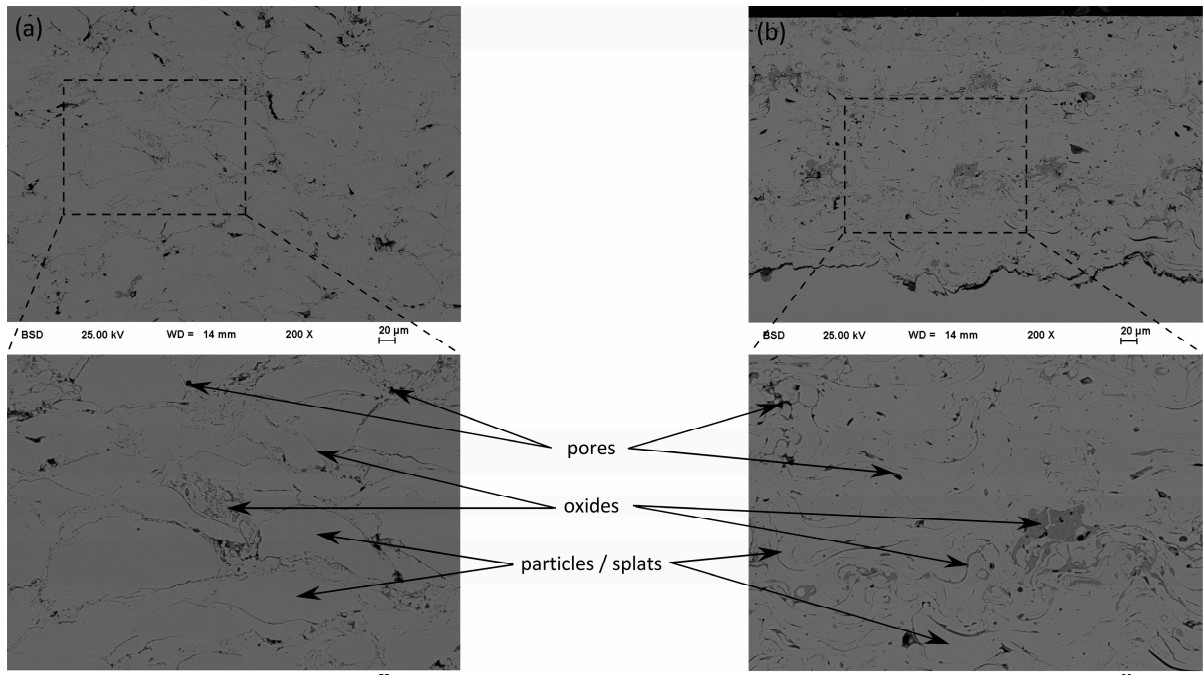

**Figure 4.** SEM images of characteristic features (pores and oxides) of AISI 316L-coatings investigated: (**a**) HVAF; (**b**) APS. The high thermal energy in the APS process causes a higher degree of melting of the particles, resulting in the typical fine lamellar structure of individual splats and larger oxidized areas.

The results of EDS measurements are presented in Figure 5, showing the average chemical composition of the samples with respect to their position on the plate. Therefore, the HVAF coating has an oxide content of about (5.1 ± 0.4) wt %, largely independent of the sample position. The APS coating has a higher average oxide content of (7.7 ± 0.6) wt %, whereby the oxide content varies more strongly within the coating and is lowest at approx. 7.0 wt % in the sample taken from the upper area.

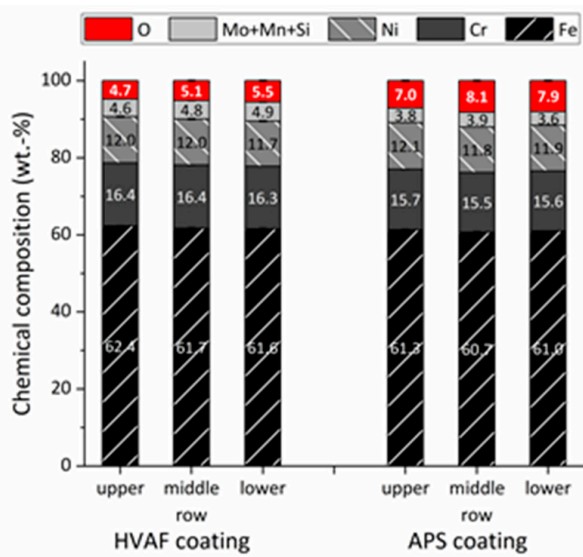

**Figure 5.** The average chemical composition of AISI 316L-coatings indicates differences in oxygen content depending on the sample position and the spraying process, with APS samples containing higher oxygen content, especially in dark colored areas.

The differences in oxide content were also determined using XRD measurements on preselected coatings (as shown in Figure 6). The patterns show that both the HVAF and APS 316L coatings contain austenite for the most part. Rietveld refinement was used to determine the following oxide contents: HVAF—middle row: 1.1 wt %, APS—upper row: 8.9 wt % and APS—middle row: 16.1 wt %.

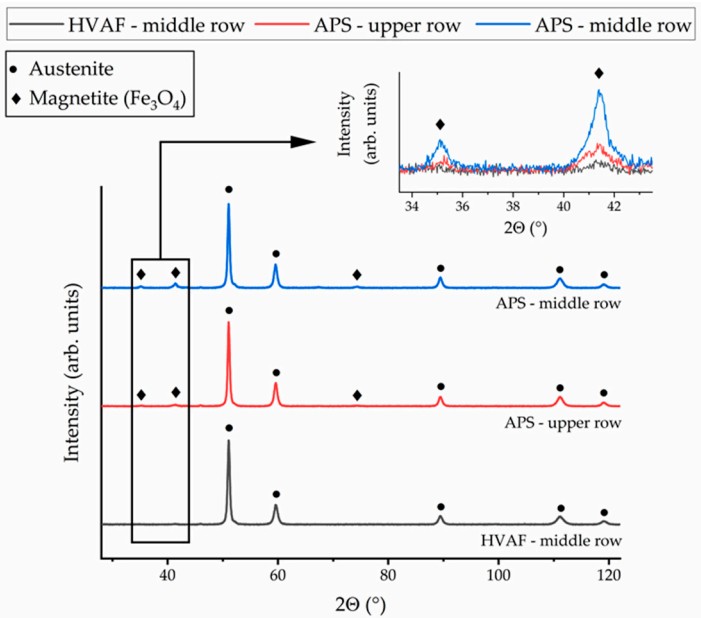

**Figure 6.** XRD patterns of selected coatings before polarization measurement reveal significant differences in oxide content (magnetite $Fe_3O_4$).

Figure 7a,b presents the polarization curves of representative selected samples of each plate. Figure 7c,d shows the corrosion current densities determined from the polarization curves in the form of a box plot with a distribution curve of the measured values next to it. The more horizontally points are positioned, the more frequently this value was measured on the sample. The polarization curves of the HVAF coating samples are very similar. Therefore, the evaluation of these polarization curves results in almost identical corrosion current densities for all samples, independent of their positioning on the plate, which is also demonstrated by the box plot and distribution in Figure 7c. The average corrosion current density of the AISI 316L HVAF coatings is $(4.5 \times 10^{-3} \pm 0.7 \times 10^{-3})$ mA/cm$^2$. Figure 7e shows the determined corrosion current density of each AISI 316L coating with corresponding position assignment on the HVAF-coated plate. Positions where no measurement was performed (middle column, middle row: due to the hole) or where the measurement was incorrect (middle column, bottom row: due to air entrapment because of insufficient filling of the gel electrolyte) are marked in black. As expected from the polarization curves, there is a uniform distribution of the corrosion current densities across the plate, so that no dependence on the sample position can be seen. The polarization curves of the AISI 316L APS coatings differ from those of the HVAF coatings. The APS samples exhibit a lower corrosion potential $E_{corr}$ and are generally at a higher current level. Especially during anodic polarization regime, the current in the APS coating samples increases significantly faster than in the HVAF coating samples, indicating a more active corrosion behavior. Only the APS samples from the top row are at a comparable current level to the HVAF coatings. Therefore, the corrosion current densities determined for the APS coating samples vary more widely and are significantly higher on average, as shown in Figure 7d. For the AISI 316L APS samples, an average corrosion current density of $(1.7 \times 10^{-2} \pm 0.8 \times 10^{-2})$ mA/cm$^2$ was determined. Figure 7f reveals the high dependence of the determined corrosion current density on the actual sample position on the plate. The samples in the upper row have especially significant lower corrosion current densities $((6.0–9.8 \times 10^{-3})$ mA/cm$^2$), compared to the plate average. The closer the tested sample is positioned to the center of the bottom row, the higher the determined corrosion current density, with a maximum of $2.9 \times 10^{-2}$ mA/cm$^2$. The determined corrosion current densities indicate a dependence on the oxide content of the investigated coating. The AISI 316L HVAF-coated plate exhibits a lower oxide content and therefore a lower corrosion current density independent of the position. The AISI 316L APS coated plate exhibits varying corrosion current densities corresponding to the local differences in oxide content. Samples with a higher oxide content (e.g., in the middle of the lower row) also have the highest corrosion current densities.

To study this correlation in more detail, the samples were investigated after measuring the polarization curves in the SEM. Figure 8 shows the top view image of selected coatings in SE and BSD contrast as well as their actual position on the plate. Since the HVAF coatings show very uniform results, only one representative coating is shown (Figure 8a). Due to the inhomogeneity, two different positions are shown for the APS-coated plate (Figure 8b,c). On the images taken with the SE detector, topographical differences become especially obvious, so that the grooves of the ground specimens and holes caused by the corrosion attack are identifiable. On the images of the sample from the HVAF coated plate (Figure 8a) and the sample from the upper row of the APS coated plate (Figure 8b), pitting can be seen only very sporadically. The corrosion attack during the polarization curves is therefore low, as suggested by the low corrosion current densities. The sample from the third row of the APS-coated plate (Figure 8c) presents significantly more pitting, correlating with increased corrosion current density. The same image segments were also taken with BSD contrast to localize the corrosion attack. The images show that pitting occurs preferentially at the oxidized regions (dark gray areas). The sample from the third row of the APS-coated plate (Figure 8c) has a significantly larger proportion of oxidized areas, leading to more pitting.

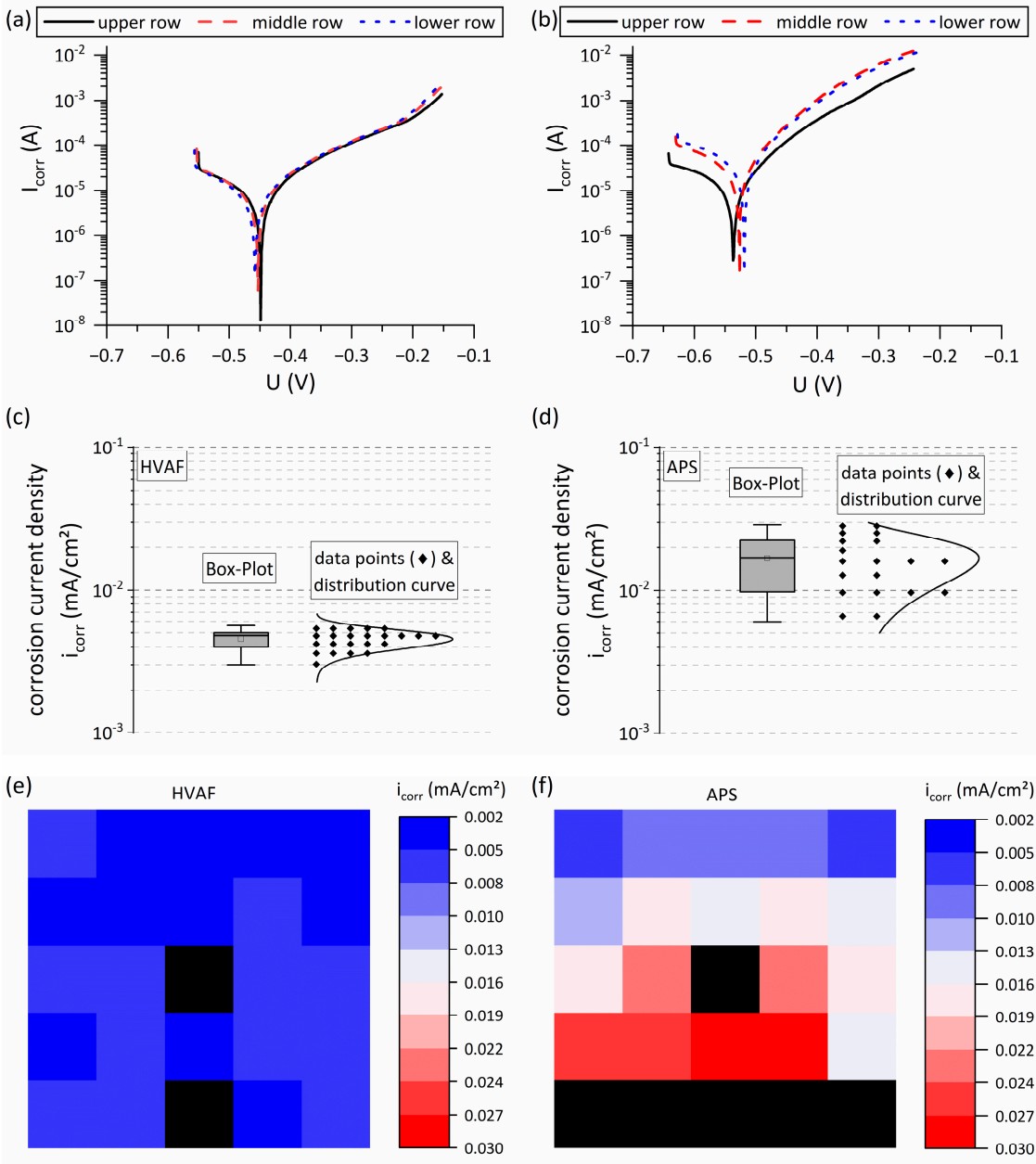

**Figure 7.** Polarization curves and corrosion current densities of the HVAF samples (**a**,**c**) are very similar regardless of location (**e**). The polarization curves and corrosion current densities of APS samples (**b**,**d**) show a different behavior (faster increase in current intensity during anodic polarization) with strong dependency to position on the plate (**f**).

Figure 9 shows the cross-sections of representative samples from the HVAF and APS-coated plate after recording the polarization curves. They clearly show that the corrosion attack takes place at the oxidized areas. In the HVAF AISI 316L coatings, the corrosion attack takes place mainly along the particle boundaries and in oxide-rich areas between individual particles. In the APS AISI 316L coatings, the oxide-rich accumulations are attacked preferentially. Based on the results, it can be seen that oxidized or oxide-rich areas are preferentially attacked in the corrosion test performed. The increased corrosion activity in oxide-rich coatings leads to increased corrosion current densities. The new test method can therefore be used to assess the quality of a coating in terms of oxide content in a quasi-non-destructive manner.

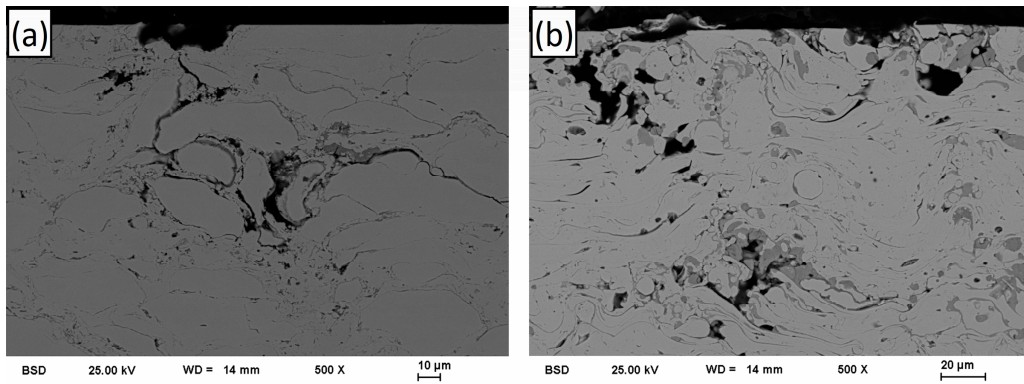

**Figure 8.** SEM images (SE and BSD contrast) of representative coatings after polarization curve measurements: (**a**) HVAF, (**b**) APS—low corrosion current density, (**c**) APS—high corrosion current density. Intensity of corrosion attack correlates with corrosion current density and occurs primarily at oxidized areas.

**Figure 9.** SEM images (BSD contrast) of cross sections of representative coatings: (**a**) HVAF; (**b**) APS. Corrosion attack occurs preferentially at oxidized areas: (**a**) HVAF—along particle boundaries; (**b**) APS—oxidized particles.

## 4. Conclusions

In this study, it was shown that polarization curves measured with an adapted measuring cell based on the three-electrode arrangement using a 3.5% NaCl gel electrolyte are suitable for identifying the different microstructures of HVAF and APS AISI 316L coatings. It can be stated that the characteristics of the APS coating led to a change in the polarization curves, resulting in higher corrosion current density and a faster increasing current as the anodic polarization progressed. While the determined corrosion current densities were constant over the entire HVAF-coated plate, large differences could be measured for the APS-coated plate as result of its more inhomogeneous microstructure.

By using the gel electrolyte, infiltration of the thermal spray coatings can be prevented, allowing the polarization curves to be recorded in a quasi-non-destructive manner and without any influence of the substrate material. The results indicate that there is a clear relationship between the determined corrosion current density and the intensity of the corrosion attack, which occurs preferentially at oxide-rich areas. Therefore, the actual method already provides the opportunity to detect difference in the oxide content of thermally sprayed AISI 316L coatings.

**Author Contributions:** Conceptualization, M.G., P.K., O.S. and C.P.; methodology, M.G., P.K., O.S. and C.P.; validation, M.G., O.S. and C.P.; investigation, M.G., O.S. and C.P.; resources, T.L. (Thomas Lampke), O.S. and C.P.; writing—original draft preparation, M.G., O.S. and C.P.; writing—review and editing, T.L. (Thomas Lindner) and T.L. (Thomas Lampke); supervision, T.L. (Thomas Lampke); project administration, T.L. (Thomas Lampke), O.S. and C.P.; funding acquisition, T.L. (Thomas Lampke), O.S. and C.P. All authors have read and agreed to the published version of the manuscript.

**Funding:** This research was funded by Arbeitsgemeinschaft industrieller Forschungsvereinigung "Otto von Guericke" e.V. (AiF) in the framework of AiF-No. ZF413911SU9, ZF4752603SU9 and ZF4820401SU9.

**Institutional Review Board Statement:** Not applicable.

**Informed Consent Statement:** Not applicable.

**Data Availability Statement:** Not applicable.

**Acknowledgments:** We would like to thank Dennis Zander, JELN Imprägnierung GmbH, for the production of the gel electrolytes and Paul Seidel and Christian Loos for their support in metallographic preparation.

**Conflicts of Interest:** The authors declare no conflict of interest. The funders had no role in the design of the study; in the collection, analyses, or interpretation of data; in the writing of the manuscript; or in the decision to publish the results.

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
