# Peer review of "Quasi Non-Destructive Quality Assessment of Thermally Sprayed AISI 316L Coatings Using Polarization Measurements in 3.5% NaCl Gel Electrolyte"

_coatings, doi:10.3390/coatings13071256_

Round 1
Reviewer 1 Report
In this manuscript a novel quasi-nondestructive technique is used to assess the corrosion performance of 316L coatings applied by HVAF and APS with the use of gel electrolyte. This work is interesting but a few aspects need to be improved. Please find a few comments below:
1) Authors refer to this technique as quasi-nondestructive. Figure 2 shows the surface of a sample after corrosion with obvious corrosion marks. Figure 8 shows the cross sections of the coating after polarization. Both figures show that the coating has been corroded and degraded. As such, this technique is not quasi-nondestructive. Authors mention it is quasi-nondestructuve since the substrate is not corroded. However, the coating is compromised and as a result I consider the term quasi-nondestructive inaccurate and potentially misleading.
2) Figure 3: The buttom of the APS sample has not been coated properly. This is also highlighted in Fig6f. Authors are kindly requested to provide an interpretation.
3) What is the mechanism behing the lower icorr values for HVAF coating over APS? Why APS shows less consictency on the icorr values as compared to HVAF? These questions need to be answered and provide a discussion in the manuscript.
4) Figure 4b: The interface of the coating/substrate appears degraded and there`s potentially delamination. Please double check.
5) Fig 6a-b: x-axis should be potential (mV). Please double check.
6) Figure 9 and lines 291-293 should be removed. Advertisement of the sequel should not be part of the conclusions.
7) It would be worth knowing the phase constitution of both coatings. XRD scans and a bried discussion would help. If authors have conducted the XRD scans in another work please provide reference.
8) This paper is focused only on icorr. Is there any data on Epit or anything that would show if this technique is applicable to measure resistance to localized forms of corrosion?
9) Authors used gel as the corrosion medium. How does this compare to the aqueous solution? Are there any discepancies? It would be worth comparing the results between the gel and 3.5% NaCl aqueous solution or at least provide a detailed explanation on the topic.
Reviewer 2 Report
The purpose of this study is to establish a non-destructive method for evaluating thermal quality of sprayed coatings. The researchers used specialized measurement techniques and gel electrolytes to identify differences in microstructures within the coatings. Additionally, this method is capable of detecting small changes in oxides contained within the coatings.
1. The method section should comprehensively describe the specific steps involved in the preparation process, including the use of gel electrolytes containing 3.5% NaCl and a 3-electrode arrangement based on a measurement pool, to ensure accurate replication of the experiment.
2. Further analysis should be conducted to evaluate the sensitivity of the measurement method to minor changes in spray parameters and raw powder materials and their impact on the measurement results.
3. The Results and Discussion section should include information on microstructural differences exhibited by coatings produced through different spray processes.
4. The above steps need to be further developed in the paper, clarifying measurement units, and providing future prospects for improving performance and availability of the measurement unit.
A comprehensive review of grammar should be conducted, including checking for spelling errors, to ensure consistency and fluency of writing style and structure throughout the entire document.
Reviewer 3 Report
The paper presents interesting results of Quasi-nondestructive polarization measurements of thermally sprayed AISI 316L coatings using in 3.5% 3 NaCl gel electrolyte. The paper is well written. The following questions need to be revised before publication:
1 The color difference between Ni and Cr elements in Figure 5 is not significant and difficult to distinguish.
2 What is the horizontal ordinate of the curve in Figures 6c and d?
Round 2
Reviewer 1 Report
The manuscript will be ready for publication after the authors remove Fig 9 (Fig 10) from the conclusions and the relevant comments. This is a scientific manuscript and as such irrelevant advertisement material needs to be removed.
Author Response
Thank you for your comment. We regret that you perceive Figure 10 and the associated descriptions as advertising. This is not our intention and we wanted to give an outlook for the measurement methodology. However, we have removed Figure 10 and the associated comments.